# Deep Leakage from Model in Federated Learning

Zihao Zhao[1], Mengen Luo[1], Wenbo Ding[1,2]

[1]Tsinghua-Berkeley Shenzhen Institute, Tsinghua Shenzhen International Graduate School

[2]Shanghai Artificial Intelligence Laboratory

{zhao-zh21, lme21}@mails.tsinghua.edu.cn, ding.wenbo@sz.tsinghua.edu.cn

Federated Learning (FL) was conceived as a secure form of distributed learning by keeping private training data local and only communicating public model gradients between clients. However, a slew of gradient leakage attacks proposed to date undermine this claim by proving its insecurity. A common limitation of these attacks is the necessity for extensive auxiliary information, such as model weights, optimizers, and certain hyperparameters (e.g., learning rate), which are challenging to acquire in practical scenarios. Furthermore, several existing algorithms, including FedAvg, circumvent the transmission of model gradients in FL by instead sending model weights, but the potential security breaches of this approach are seldom considered. In this paper, we propose two innovative frameworks, DLM and DLM+, that reveal the potential leakage of private local data of clients when transmitting model weights under the FL framework. We also conduct a series of experiments to elucidate the impact and universality of our attack frameworks. Additionally, we propose and evaluate two defenses against the proposed attacks, assessing their protective efficacy.

## 1. Introduction

The explosive growth and increasing complexity of the data have raised huge difficulties as well as challenges to the traditional centralized machine learning schemes due to their heavy dependency on the local high-quality computing and storage resources. In this context, the *distributed learning* [1] has emerged as an effective solution to utilize the ubiquitous but low-quality resources to tackle the large-scale data problem, especially in the era of the Internet of things and edge computing. One realistic application of distributed learning is federated learning (FL) [2], which intends to keep clients' training data locally rather than transmitting them to others to protect the privacy of each client. Specifically, during the training process, all clients train their models locally using their own private data, calculate the model update information (such as model weights and model gradients) and then upload them to the parameter server, which will aggregate this information to update the global model. Thanks to its privacy protection capability, FL has been successfully applied in the financial and medical fields [3].

However, even without uploading plain training data to the parameter server in FL, some security problems still exist in such procedures such as the **data leakage problem**. Recently, [4] presented a security breach for the gradient transmitting frameworks called deep leakage from gradients (DLG) to recover each client's immovable training data, which gives us a new insight into attacking the clients' private data from gradients in FL. For instance, [5] utilized the sign of CrossEntropy loss to estimate the ground-truth label and increase the accuracy of DLG. However, it could only be effective with clients who use CrossEntropy loss rather than other loss functions. [6] made use of cosine similarity to measure the distance between the dummy gradient and ground-truth gradient rather than the mean square error (MSE). [7] attempted to recover the batch input before the fully connected layer by solving the linear equation. However, strong assumptions are made to solve the equations, and data recovery cannot be guaranteed under more general conditions. [8] proposed an image batch restoration approach called GradInversion by matching gradients while regularizing image fidelity.

Unfortunately, these restoration approaches require not only model gradients transmitted in FL, but also some auxiliary information, such as model parameters, the optimizer, and some hyperparameters (e.g. learning rate), which are difficult to obtain in the practice. Moreover, by realizing the hazards of uploading gradients of models, a number of recent FL frameworks turn to transmitting model parameters to perform server aggregation and model updating such as FedAvg [9], which have not proven to be attackable directly. Nevertheless, we recently found that the transmission of model weights is also insecure and assaultable. In this paper, we intend to steal the private training data of clients by the communicated model parameters in FL, which brings a huge challenge to the foundation of FL. In particular, we mainly address two challenges: a) how to design an innovative loss function to enable the adversary to recover the ground-truth data using model parameters since the loss function of all the gradient leakage attacks cannot be utilized directly on the model parameter leakage situation, and b) how to apply this loss function to FL scenarios. Our attempt to tackle these two challenges leads to two novel attack frameworks, which we call the **D**eep **L**eakage from **M**odel (DLM) and DLM+ for recovering the private training data of clients. The main contributions of our work are summarized as threefold:

1. We identify the possibility of recovering private training data from each client in FL by utilizing only the transmitted model parameters and loss function without any prior knowledge of the training data, which poses an unprecedented challenge to the foundation of FL.

2. We present two novel frameworks DLM and DLM+ based on two innovative loss functions and apply them to the FedAvg, which is the most famous and widely-used algorithm in FL. In the experiment part, the result illustrates that the FL architectures that exchange model weights between the server and clients could not manage to protect the private data of clients.

3. A variety of experiments are conducted to compare different deep gradient leakage approaches and the results demonstrate that our proposed model leakage attacks obtain more accurate results compared with existing gradient leakage attacks, and we also give an intuitive explanation towards it. Moreover, two defenses are introduced to protect the FL system from the model leakage attack.

## 2. Preliminaries

In this section, we will introduce some preliminaries about FL and the specific process of a widely used algorithm in FL called FedAvg. Additionally, the threat model of this literature is presented, which illustrates what the adversary could obtain and how our attack takes effect.

### 2.1. Federated Learning

FL, first proposed by Google in 2017 [9], can be regarded as a distributed machine learning framework that is able to provide private local data protection. During the entire FL training process, the private data of each clientare not only kept unknown by the server but also invisible to other clients participating in the training. After completing the training process, all clients build a global shared model to realize implicit data sharing and win-win cooperation.

Specifically, for an FL paradigm that transmits model weights such as FedAvg [9], assume that the system contains $K$ local clients $\{C_1, C_2, \cdots, C_K\}$ and each client has the dataset $D_k = \left\{ \left( \boldsymbol{x}^{(i)}, y^{(i)} \right) \right\}_{i=1}^{n_k}$, where $k \in \{1 \ldots K\}$, and $n_k = |D_k|$ is the number of training samples in $D_k$. During the training process, $K$ clients jointly train a global shared model with the help of the global parameter server without exposing their local data. At the beginning of iteration $t$, the parameter server sends the global model $\boldsymbol{W}_g^t$ to each client, and each client $k$ trains it with his own dataset $D_k$ and updates the global model $\boldsymbol{W}_g^t$ by $\boldsymbol{W}_k^{t+1} \leftarrow \boldsymbol{W}_g^t - \alpha \nabla \boldsymbol{W}_k^t$, where $\alpha$ is the local learning rate, then uploads the $\boldsymbol{W}_k^{t+1}$ to the global server. Subsequently, the server will update the global parameter by $\boldsymbol{W}_g^{t+1} \leftarrow \frac{1}{K} \sum_{k=1}^{K} \boldsymbol{W}_k^{t+1}$ for iteration $t + 1$.

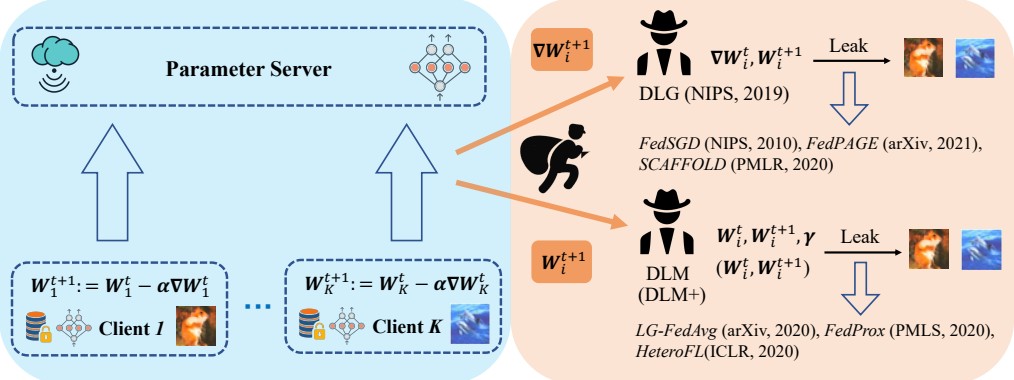

Figure 1: The frameworks of the proposed DLM and DLM+.

## 2.2. Threat Model

In this work, the goal of our attack is to recover the private local data of each client in an FL system only by utilizing communicated model parameters. According to previous work on data leakage from gradients, e.g., [4], suppose that the adversary is aware of the client model parameters, optimizers, and some hyperparameters such as the learning rate, which are not usually broadcast in most distributed learning frameworks and are arduous to obtain for adversaries. In our threat model, in contrast, the only two factors that the adversary needs to gain are the weights of the transmitted model and the loss function of each client. To this end, to recover the private data, the adversary first utilizes the procured model weights to construct a neural network similar to the clients. Thereafter, a pair of randomly initialized dummy data and labels are placed in the constructed neural network. With the loss function of each client, the corresponding dummy model gradients can be calculated. Most importantly, the adversary employs the dummy model gradients to compute the innovative loss function presented in Sections 3 & 4, which does not contain any prior knowledge of training data. When the algorithm converges successfully, the dummy data and dummy label will become rather close to the ground-truth data and label by approximating the true model update.

# 3. Deep Leakage from Model (DLM)

In this section, the definition of a distributed learning system derived from real-world application scenarios is first proposed. According to this system, we formulate our novel model attack framework and provide a vanilla attack to FedAvg to recover the private data.

## 3.1. Model Updating in Local Systems

As shown in the left part of Figure 1, at iteration $t+1$, each local client employs the common stochastic gradient descent (SGD) method [10] to update the model weight by:

$$W^{t+1} := W^t - \alpha \nabla W^t, \tag{1}$$

where $W^t$ and $W^{t+1}$ are the model weights of a local client at iteration $t$ and $t+1$ respectively, $\alpha$ is the learning rate and $\nabla W^t$ is the model gradients.

The above Equation (1) is a general formula of SGD, but we need a more detailed formula in the following derivation. Assume $(x^*, y^*)$ is the ground-truth data and label, $\mathcal{L}(\cdot)$ is the local loss function and the $\nabla_{W^t} \mathcal{L}(x^*, y^*)$ is the loss w.r.t. model weights $W^t$, the Equation (1) could also be written as:

$$W^{t+1} := W^t - \alpha \nabla_{W^t} \mathcal{L}(x^*, y^*),$$
$$\nabla_{W^t} \mathcal{L}(x^*, y^*) = \nabla W^t = \frac{\partial \mathcal{L}(\mathcal{F}(x^*, W^t), y^*)}{\partial W^t}, \tag{2}$$

where the shared neural network function is indicated as a differential model $\mathcal{F}(\cdot)$.

In addition to SGD, batch gradient descent (BGD) [11] and mini-batch gradient descent (MBGD) [12] are also frequently used in distributed learning as optimizers with multiple local iterations. However, in the main part of this paper, we just analyze the SGD situation, and the MBGD with multiple local iterations will be discussed in the Supp. material part.

## 3.2. Adversary Attacks

In the FL attack process, at the beginning of the iteration $t$, the adversary can intercept the global model $\boldsymbol{W}_g^t$ transmitted from the parameter server to clients. After an arbitrary client in the FL system finishes its local training process, the adversary can gain the model weight $\boldsymbol{W}_k^{t+1}$ of the client $k$ uploading to the server. After obtaining all this information, the attacker can steal the local data from the transmitted model weights as shown in the right part of Figure 1. Specifically, on the attacker side, it is assumed that the adversary knows model weights at each iteration and is unknown to other information, such as the client's learning rate $\alpha$ and model gradients $\nabla \boldsymbol{W}$.

Similar to the DLG, the attacker in our approach first randomly initialize a dummy data input $\hat{\boldsymbol{x}}$ and dummy label $\hat{y}$. Since the attacker knows the global model weights $\boldsymbol{W}_g^t$ in FL, one is able to put $\hat{\boldsymbol{x}}$ and $\hat{y}$ into the model $\boldsymbol{W}_g^t$ and calculate a dummy gradient $\nabla_{\boldsymbol{W}_g^t} \mathcal{L}(\hat{\boldsymbol{x}}, \hat{y})$ using the same loss function $\mathcal{L}(\cdot)$ as each client. Subsequently, the adversary is able to restore the client $k$'s private data by minimizing the distance between the plain gradient and dummy gradient by the following naive *objective function*:

$$\mathcal{L}_{DLM}(\hat{\boldsymbol{x}}, \hat{y}, \gamma) = \| \nabla_{\boldsymbol{W}_g^t} \mathcal{L}(\hat{\boldsymbol{x}}, \hat{y}) - \gamma * \left( \boldsymbol{W}_k^t - \boldsymbol{W}_k^{t+1} \right) \|_F^2, \tag{3}$$

where $\gamma$ is a tuning factor to compensate for the effect of the learning rate $\alpha$ of each client and makes the term $\gamma * \left( \boldsymbol{W}_k^t - \boldsymbol{W}_k^{t+1} \right)$ approximate the ground-truth gradient. Thereafter, the adversary utilizes the gradient of this objective function with respect to its dummy data input $\nabla_{\hat{\boldsymbol{x}}_i} \mathcal{L}_{DLM}(\hat{\boldsymbol{x}}, \hat{y}, \gamma)$ to update $\hat{\boldsymbol{x}}$. Similarly, the dummy label $\hat{y}$ and the tuning factor $\gamma$ are updated by $\nabla_{\hat{y}_i} \mathcal{L}_{DLM}(\hat{\boldsymbol{x}}, \hat{y}, \gamma)$ and $\nabla_{\gamma_i} \mathcal{L}_{DLM}(\hat{\boldsymbol{x}}, \hat{y}, \gamma)$, respectively.

Nevertheless, this approach has some drawbacks. At first, this algorithm has to update three mutually-dependent variables step by step, in which these three decision variables can potentially affect each other and result in high requirements for initialization. Second, it introduces too much randomness in the training process and increases the difficulty of model convergence, which needs to be optimized further.

# 4. Deep Leakage from Model+ (DLM+)

In this section, we propose a stronger attack framework called Deep Model Leakage+ to restore the training data of each client more stably and precisely. First, the expurgation of the learning rate $\alpha$ from Equation (2) is performed, which decreases the randomness of the algorithm and obtains a better convergence. Then, we present a new objective function without the appearance of $\alpha$ and show how it works in FedAvg.

## 4.1. Expurgate the learning rate $\alpha$

According to Equation (1), we take the $Frobenius$ norm [13] on both sides of it:

$$\| \boldsymbol{W}^t - \boldsymbol{W}^{t+1} \|_F = \alpha \| \nabla_{\boldsymbol{W}^t} \mathcal{L}(\boldsymbol{x}^*, y^*) \|_F. \tag{4}$$

Hence, we get the expression of $\alpha$ as follows:

$$\alpha = \frac{\| \boldsymbol{W}^t - \boldsymbol{W}^{t+1} \|_F}{\| \nabla_{\boldsymbol{W}^t} \mathcal{L}(\boldsymbol{x}^*, y^*) \|_F}, \tag{5}$$

where the denominator $\| \nabla_{\boldsymbol{W}^t} \mathcal{L}(\boldsymbol{x}^*, y^*) \|_F$ is unknown to the adversary. Afterwards, plugging Equation (5) back into the model update expression, Equation (1), and will give the following

expression only related to model weights at two iterations $\boldsymbol{W}^t$, $\boldsymbol{W}^{t+1}$ and the gradient of the loss w.r.t $\boldsymbol{W}^t$:

$$\frac{\boldsymbol{W}^t - \boldsymbol{W}^{t+1}}{\|\boldsymbol{W}^t - \boldsymbol{W}^{t+1}\|_F} = \frac{\nabla_{\boldsymbol{W}^t}\mathcal{L}(\boldsymbol{x}^*, y^*)}{\|\nabla_{\boldsymbol{W}^t}\mathcal{L}(\boldsymbol{x}^*, y^*)\|_F}. \tag{6}$$

Note that the adversary can directly compute $\|\nabla_{\boldsymbol{W}^t}\mathcal{L}(\boldsymbol{x}^*, y^*)\|_F$ once the model parameter $\boldsymbol{W}^t$ is obtained. Compared with Equation (2), the new equation Equation (6) without $\alpha$ reduces the randomness of the algorithm and hence gains better performance.

## 4.2. Adversary Attacks

On the attacker side, for each client $k$ in FL, we introduce our innovative *objective function* $\mathcal{L}_{DLM+}(\hat{\boldsymbol{x}}, \hat{y})$, which does not have an item related to the learning rate $\alpha$:

$$\mathcal{L}_{DLM+}(\hat{\boldsymbol{x}}, \hat{y}) = \left\| \frac{\nabla_{\boldsymbol{W}_k^t}\mathcal{L}(\hat{\boldsymbol{x}}, \hat{y})}{\|\nabla_{\boldsymbol{W}_k^t}\mathcal{L}(\hat{\boldsymbol{x}}, \hat{y})\|_F} - \frac{\boldsymbol{W}_g^t - \boldsymbol{W}_k^{t+1}}{\|\boldsymbol{W}_g^t - \boldsymbol{W}_k^{t+1}\|_F} \right\|_F^2. \tag{7}$$

Afterward, the adversary uses the following equation to update the dummy data $\hat{\boldsymbol{x}}$ and the dummy label $\hat{y}$:

$$\hat{\boldsymbol{x}} \leftarrow \hat{\boldsymbol{x}} - \eta\nabla_{\hat{\boldsymbol{x}}}\mathcal{L}_{DLM+}(\hat{\boldsymbol{x}}, \hat{y})$$
$$\hat{y} \leftarrow \hat{y} - \eta\nabla\hat{y}\mathcal{L}_{DLM+}(\hat{\boldsymbol{x}}, \hat{y}). \tag{8}$$

The full algorithm is shown as **Algorithm 2**. In this algorithm, the private data of each client are always kept locally, which completely obeys the rules of FL. However, there is still a probability of restoring the private data by using the transmitted model weights, which raises a huge challenge to the security of FL.

# 5. Experiments

**Setup.** The proposed frameworks are evaluated with the MNIST, CIFAR10, and CIFAR100 datasets to cover different sizes of images. In our FL system, 10 clients are employed with 10% IID partition of the total dataset. To ensure the generalization of our frameworks, 2 widely adopted models (MLP and LeNet) are selected as the attack model. On the adversary part, LBFGS [14] is employed for optimization to LeNet with learning rate $\eta = 1$, history size = 100, number of iterations = 200 and CrossEntropy loss function. For the three-layer MLP network, Adam [15] with learning rate $\eta = 0.1$ works as the optimizer, and the number of iterations = 4000 and the CrossEntropy loss function are selected while training the model.

**Evaluation Metrics.** To measure the similarity between the restoration results and the ground-truth image, we choose three popular visual comparisons of images to evaluate the pixel-wise discrepancies: i) the peak signal-to-noise ratio (PSNR) [16] to calculate the error between pixels, ii) the learned perceptual image patch similarity (LPIPS) [17] to measure image similarity with deep features, and iii) the structural similarity (SSIM) [18] to measure the correlation, luminance distortion and contrast distortion of images.

## 5.1. Comparison Results

First, this experiment compares the three proposed frameworks with two gradient leakage approaches as follows.

(a) DLG [4]: This deep leakage approach utilizes the $\ell_2$ distance between ground-truth gradients and dummy gradients as the objective function to update the dummy data and dummy labels.

(b) Cosine similarity [6]: This approach designs an objective function based on angles between ground-truth gradients and dummy gradients, i.e., cosine similarity, to restore the ground truth image and label.

In particular, fifty different images from three datasets are chosen to acquire the average test accuracy, PSNR, LPIPS, and SSIM values. For fair comparisons, the iterations and models for all methods are the same. In our experiment, if the PSNR of one test is more than 30 dB, this test will be considered a successful recovery. Based on this condition, the restoration results are shown in Table 1, where our framework DLM+ has the best accuracy, PSNR, and SSIM values among other algorithms. This indicates that the restoration result of DLM+ is the greatest and that its stability is the best. As for DLM, it has only a moderate accuracy and PSNR due to the randomness caused by the one extra decision variable $\gamma$ and tough convergence, but it achieves the highest LPIPS value, which demonstrates that DLM is superior in the measurement of deep features of image similarity.

Table 1: Restoration results of different algorithms under FL scenarios.

| Algorithm | Acc↑ | PSNR↑ | LPIPS ↓ | SSIM ↑ |
|---|---|---|---|---|
| DLG | 64% | 39.99 | 1.17*1E-4 | 0.68 |
| Cosine | 90% | 32.87 | 4.84*1E-3 | 0.91 |
| DLM | 68% | 41.98 | **4.74*1E-6** | 0.64 |
| **DLM+** | **92%** | **46.96** | 1.02*1E-4 | **0.92** |

**Why DLM is better than DLG?** To illustrate the reason, we slightly modify the objective function of the original DLG by adding a tuning factor $k$, which is similar to the form of DLM:

$$\mathcal{L}'_{DLG}(\hat{\boldsymbol{x}}, \hat{y}, k) = \left\| \nabla_{\boldsymbol{W}_g^t} \mathcal{L}(\hat{\boldsymbol{x}}, \hat{y}) - k * \nabla \boldsymbol{W}_k^t \right\|_F^2 , \tag{9}$$

where $k = 1$ will induce the original DLG. Since the original DLG utilizes the ground-truth gradient, it can converge faster but has less range to search for the optimal solution. Adding the tuning factor can appropriately enlarge the searching margin and increase the probability to recover the original data. The simulation results with 100 repeated experiments are summarized in Table 2.

Table 2: DLG with different tuning factors $k$.

| $k =$ | 1(baseline) | 0.9 | 0.95 | 1.05 | 1.1 |
|---|---|---|---|---|---|
| ACC | 67% | 82% | 79% | 82% | 85% |
| PSNR | 44.29 | 52.79 | 51.03 | 52.73 | 54.29 |
| LPIPS | 2.78e-6 | 5.90e-6 | 5.33e-6 | 1.20e-3 | 1.1e-3 |
| SSIM | 0.6699 | 0.8199 | 0.7899 | 0.81999 | 0.8499 |

The above results validate the effectiveness of introducing the tuning factor $k$ to DLG, in terms of both accuracy and restoration quality. What's more, we could rewrite the objective function of DLM exactly similar to Equation (9):

$$\begin{aligned} \mathcal{L}_{DLM}(\hat{\boldsymbol{x}}, \hat{y}, \gamma) &= \left\| \nabla_{\boldsymbol{W}_g^t} \mathcal{L}(\hat{\boldsymbol{x}}, \hat{y}) - \gamma * \left( \boldsymbol{W}_k^t - \boldsymbol{W}_k^{t+1} \right) \right\|_F^2 \\ &= \left\| \nabla_{\boldsymbol{W}_g^t} \mathcal{L}(\hat{\boldsymbol{x}}, \hat{y}) - \gamma * \alpha * \nabla \boldsymbol{W}_k^t \right\|_F^2 \\ &= \mathcal{L}'_{DLG}(\hat{\boldsymbol{x}}, \hat{y}, \gamma * \alpha). \end{aligned} \tag{10}$$

Therefore, such improvements give us an intuitive explanation for why the DLM performs better than the original DLG. More results can be found in Appendix A.3. Additionally, we also conduct some evaluations on batch data in Appendix B.

## 5.2. The Influence of $\gamma$'s Initial Value

In the DLM framework, the adversary has to randomly choose an initial value of the tuning factor $\gamma$. However, the choice of the initial value can greatly affect the convergence and precision of the algorithm. For the experimental setting, since the local client takes the learning rate $\alpha = 0.01$, the

$\gamma$ is supposed to be $1/\alpha = 100$ in theory. Therefore, different values of $\gamma$ near 100 are chosen to investigate the impact of the tuning factor on both the LeNet and MLP networks.

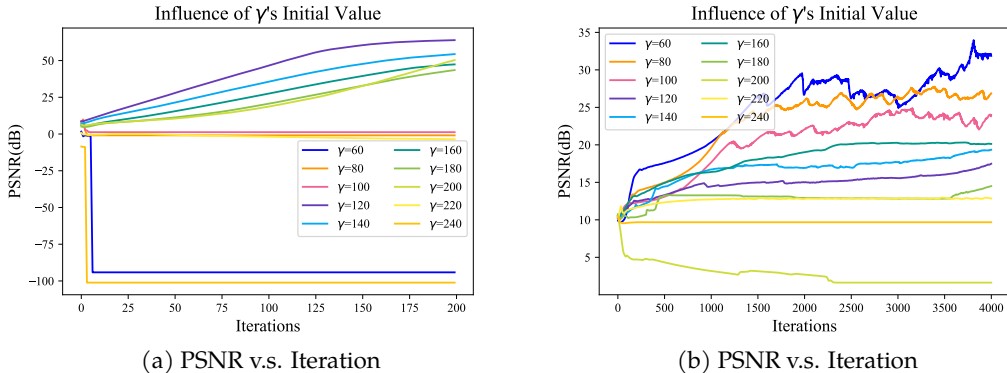

(a) PSNR v.s. Iteration                    (b) PSNR v.s. Iteration

Figure 2: Performance (PSNR) comparison of (a) LeNets and (b) MLP Network with various $\gamma$'s initial value.

Figure 2a shows that when the $\gamma$ is lower than or equal to the ideal value 100, the result of restoration is completely unrecognizable. With the growth of $\gamma$, the PSNR reaches the peak at $\gamma = 120$, and then starts decreasing until $\gamma = 240$ and the PSNR almost drops to 0. As for the MLP network, since the it utilizes the Adam as the optimizer, it achieves the highest PSNR at $\gamma = 60$ and then decreases, as shown in Figure 2b.

These two figures illustrate that the DLM is difficult to find a quite precise value $\gamma$ to obtain a high PSNR restoration result, because if clients adopt different learning rates $\alpha$ or different optimizers, the optimal value of $\gamma$ will change. However, in the DLM+, there is no need to initialize any extra hyperparameter and thus the randomness of the framework is reduced.

## 5.3. Local Iteration & Momentum

In the basic experimental setting, we consider only the local training process with 1 local epoch in DLM+. Nevertheless, in practical situations, each local client will execute several training epochs during the local training time, or sometimes the adversary may not be able to intercept the continuous model parameters $\boldsymbol{W}^t$ and $\boldsymbol{W}^{t+1}$ but only a few rounds of model parameters in between (i.e., $\boldsymbol{W}^t$ and $\boldsymbol{W}^{t+k}, k > 1$). Hence, different local epochs are considered to evaluate the robustness of our architectures. As Figure 3a shows, for the LeNet, the PSNR of the restoration image after 100 local epochs is still almost the same in the basic experiment with 1 local step.

In addition, we also change the optimizer for local clients. Prior experiments mainly focus on the vanilla SGD, whereas SGD with momentum is more widely used to accumulate historical gradient information momentum to accelerate the vanilla SGD at present. Figure 3b indicates that the performance of DLM+ in SGD with momentum is as good as vanilla SGD.

## 5.4. Defenses

In this section, we introduce two defense approaches: differential privacy and model sparsification to protect the client from our proposed attacks.

**Differential privacy.** One protection method is to train a neural network under differential privacy (DP) [19]. Instead of applying DP to the model gradients [20], we turn to performing DP on the model weights. Specifically, after finishing the local training process and updating the model weights, each $l$-th layer $\boldsymbol{W}_{(l)}^t$ of model weights is clipped as follows:

$$\overline{\boldsymbol{W}}_{(l)}^t \leftarrow \boldsymbol{W}_{(l)}^t / \max\left(1, \frac{\|\boldsymbol{W}_{(l)}^t\|_2}{C}\right), \tag{11}$$

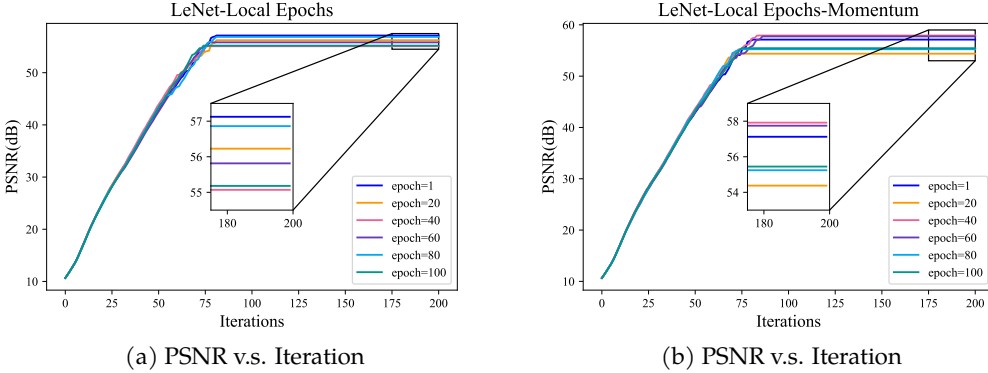

(a) PSNR v.s. Iteration  (b) PSNR v.s. Iteration

Figure 3: Performance (PSNR) comparison of LeNet with (a) SGD and (b) SGD with momentum under different local iterations.

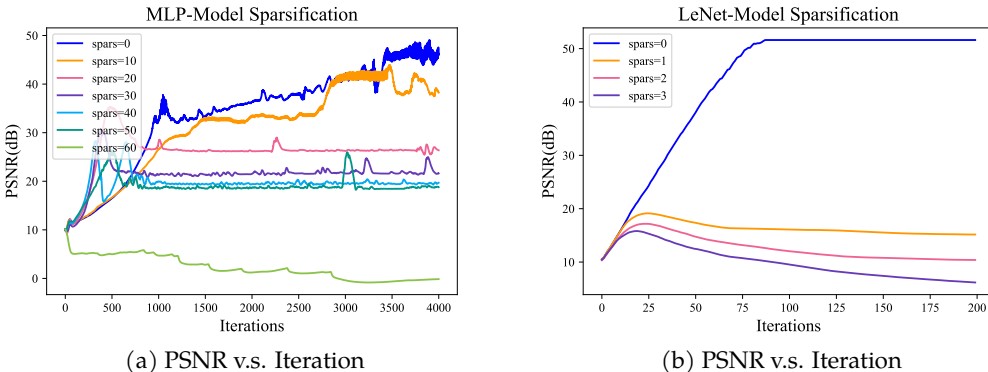

(a) PSNR v.s. Iteration  (b) PSNR v.s. Iteration

Figure 4: Performance (PSNR) comparison of applying model sparsification on (a) MLP network and (b) LeNet.

where $\overline{\boldsymbol{W}}_{(l)}^{t}$ is the clipped model parameter and $C$ is a clipping threshold. Subsequently, the noise of different strengths is added to the clipped model weights for better evaluation, including Gaussian noise and Laplacian noise with variance ranging from $10^{-1}$ to $10^{-5}$ and central 0:

$$\widetilde{\boldsymbol{W}}_{(l)}^{t} \leftarrow \frac{1}{L} \left( \overline{\boldsymbol{W}_{(l)}^{t}} + N \left( 0, \sigma^2 C^2 \boldsymbol{I} \right) \right), \tag{12}$$

where $L$ is the group size, $C$ is the model parameter norm bound and $\sigma$ is the noise level. After employing DP with Gaussian noise between 0 and 1e-02, the PSNR results of Gaussian DP are shown in Section 5.4. When the strength grows up to more than 1e-04, our attack algorithms fail to recover the training data. Therefore, the recommended strength of Gaussian noise is 1e-04. In addition, the results of LeNet and Laplacian noise are shown in Section 5.4, which demonstrates a trend similar to that of the above result.

The above results indicate that only a relatively low level of noise should be added to the communicated model parameters to prevent the whole FL system from our attacks. The reason is that a strong noise may severely disturb the parameters of the original model parameters $\boldsymbol{W}_g^t$ and $\boldsymbol{W}_k^{t+1}$.

**Model Sparsification.** Model sparsification [21] is also an effective method to prevent clients from attacks. Particularly, before uploading their model weights to the parameter server, the clients execute the model sparsification range from 0 to 60%, in which it first sets a quantile of the whole model weight, then if the weight value is less than the chosen quantile, it will be pruned to zero. Figure 4 illustrates that when the sparsification rate is approximately 20%, DLM+ still obtains a PSNR restoration result greater than the value of 35. With the increases in sparsities, the PSNR of DLM+ keeps decreasing but DLM+ could still manage to recover identifiable training data when the sparsification rate is between 1% and 50%. Once the sparsity level exceeds the tolerance of DLM+

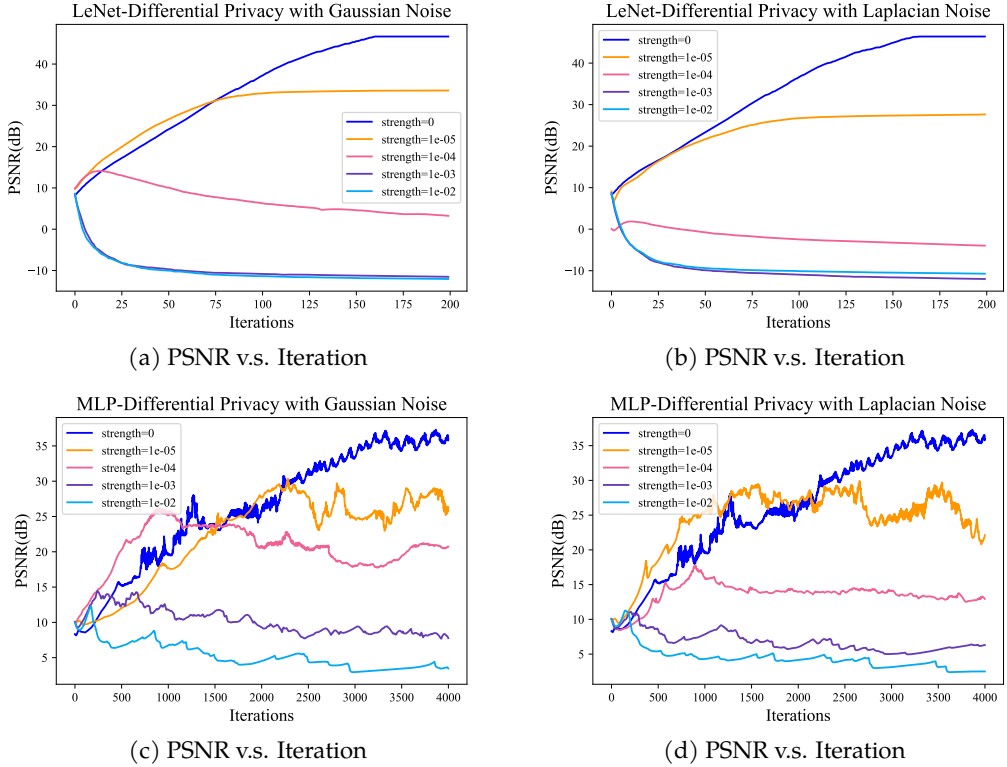

(a) PSNR v.s. Iteration

(b) PSNR v.s. Iteration

(c) PSNR v.s. Iteration

(d) PSNR v.s. Iteration

Figure 5: Performance (PSNR) comparison of applying differential privacy with (a)/(c) various Gaussian noise (b)/(d) various Laplacian noise on LeNet/MLP network.

around 50%, it is difficult for the adversary model to find the ground-truth gradient and the PSNR becomes quite low. Therefore, a model sparsification scale larger than 50% of model sparsification could be sufficient to protect the transmitted model weights in the MLP network.

However, Figure 4 also indicates that the resistance of the DLM+ attack to model sparsification is extremely poor, with a sharply shrunken PSNR even when the sparsity of LeNet is only 1.

## 6. Conclusion and Discussion

In this paper, we present two novel frameworks: Deep Leakage from Model (DLM) and Deep Leakage from Model+ (DLM+) to recover the private data of each client. The experiment shows that our proposed frameworks achieve the highest accuracy, PSNR, and SSIM value compared with existing gradient leakage approaches, including DLG and cosine similarity. For a better comparison, we also add a tuning factor to the original DLG to explain why DLM outperforms the DLG. In addition, the analysis of the impact of the local iteration and local optimizer verifies that both of our proposed algorithms are quite robust to multiple local iterations. Finally, two defense methods to protect the clients from our attacks are evaluated, and the corresponding protection thresholds are recommended.

**Discussion.** In the above sections, we mainly discuss the attacks in federated learning since the FL frameworks are specially designed to protect the security of private data. However, we have to emphasize two points: (a) the DLM and DLM+ could still work on the non-IID and a number of clients' circumstances since the attacking process is similar, (b) our two proposed architectures could still steal the private training data from these distributed learning frameworks that transmitted model parameters, such as distributed SGD [22].

# Acknowledgements

This work was supported by the National Key R&D Program of China under Grant No.2022ZD0160504, by Shenzhen Ubiquitous Data Enabling Key Lab under Grant ZDSYS20220527171406015, and by Tsinghua Shenzhen International Graduate School-Shenzhen Pengrui Young Faculty Program of Shenzhen Pengrui Foundation (No. SZPR2023005). We would also like to thank anonymous reviewers for their insightful comments.

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

# A. Appendix

The section contains supplementary extensional algorithms and experimental results.

## A.1. Frameworks for the distribution learning and DLM

First of all, as we mentioned in the discussion part, our two proposed architectures could also be applied to the these distributed learning (DL) frameworks which transmitted model parameters and obtain their private training data.

---

**Algorithm 1** DLM for FedAvg

---

**Input:** The $K$ clients are indexed by $k$; $\boldsymbol{W}_k^t$ and $\boldsymbol{W}_k^{t+1}$ are the model weight of client $k$ at iteration $t$ and $t+1$ respectively, $\boldsymbol{W}_g^t$ is the global model weight at iteration $t$, $N$ is number of iterations and $\eta$ is the attacker's learning rate
**function** DLM for FedAvg
  **At iteration** $t+1$**:**
  **for** each client $k$, the adversary **do**
    The adversary knows the value of $\boldsymbol{W}_g^t$ and $\boldsymbol{W}_k^{t+1}$
    Randomly initialize dummy data $\hat{\boldsymbol{x}} \leftarrow \mathcal{N}(0, \boldsymbol{I})$ and dummy label $\hat{y} \leftarrow \mathcal{N}(0, 1)$
    **for** $i = 1$ **to** $N$ **do**
$$\nabla_{\boldsymbol{W}_g^t} \mathcal{L}(\hat{\boldsymbol{x}}_i, \hat{y}_i) = \frac{\partial \mathcal{L}\left(\mathcal{F}\left(\hat{\boldsymbol{x}}_i, \boldsymbol{W}_g^t\right), \hat{y}_i\right)}{\partial \boldsymbol{W}_g^t}$$
      Calculate the loss $\mathcal{L}_{DLM}(\hat{\boldsymbol{x}}_i, \hat{y}_i, \gamma_i)$ by Equation (3)
      $\hat{\boldsymbol{x}}_{i+1} \leftarrow \hat{\boldsymbol{x}}_i - \eta \nabla_{\hat{\boldsymbol{x}}_i} \mathcal{L}_{DLM}(\hat{\boldsymbol{x}}_i, \hat{y}_i, \gamma_i)$
      $\hat{y}_{i+1} \leftarrow \hat{y}_i - \eta \nabla_{\hat{y}_i} \mathcal{L}_{DLM}(\hat{\boldsymbol{x}}_i, \hat{y}_i, \gamma_i)$
      $\gamma_{i+1} \leftarrow \gamma_i - \eta \nabla_{\gamma_i} \mathcal{L}_{DLM}(\hat{\boldsymbol{x}}_i, \hat{y}_i, \gamma_i)$
    **end for**
  **end for**
**end function**

---

---

**Algorithm 2** DLM+ for FedAvg

---

**Input:** The $K$ clients are indexed by $k$; $\boldsymbol{W}_k^t$ and $\boldsymbol{W}_k^{t+1}$ are the model weight of client $k$ at iteration $t$ and $t+1$ respectively, $\boldsymbol{W}_g^t$ is the global model weight at iteration $t$, $N$ is number of iterations and $\eta$ is the attacker's learning rate
**function** DLM+ for FedAvg
  **At iteration** $t+1$**:**
  **for** each client $k$, the adversary **do**
    The adversary knows the value of $\boldsymbol{W}_g^t$ and $\boldsymbol{W}_k^{t+1}$
    Randomly initialize dummy data $\hat{\boldsymbol{x}} \leftarrow \mathcal{N}(0, \boldsymbol{I})$ and dummy label $\hat{y} \leftarrow \mathcal{N}(0, 1)$
    **for** $i = 1$ **to** $N$ **do**
$$\nabla_{\boldsymbol{W}_g^t} \mathcal{L}(\hat{\boldsymbol{x}}_i, \hat{y}_i) = \frac{\partial \mathcal{L}\left(\mathcal{F}\left(\hat{\boldsymbol{x}}_i, \boldsymbol{W}_g^t\right), \hat{y}_i\right)}{\partial \boldsymbol{W}_g^t}$$
      Calculate the loss $\mathcal{L}_{DLM+}(\hat{\boldsymbol{x}}_i, \hat{y}_i)$ by Equation (7)
      $\hat{\boldsymbol{x}}_{i+1} \leftarrow \hat{\boldsymbol{x}}_i - \eta \nabla_{\hat{\boldsymbol{x}}_i} \mathcal{L}_{DLM+}(\hat{\boldsymbol{x}}_i, \hat{y}_i)$
      $\hat{y}_{i+1} \leftarrow \hat{y}_i - \eta \nabla_{\hat{y}_i} \mathcal{L}_{DLM+}(\hat{\boldsymbol{x}}_i, \hat{y}_i)$
    **end for**
  **end for**
**end function**

---

## A.2. Model Setting

In this section, we introduce the structure of our experiment models (MLP and LeNet) in Table 3 and Table 4 respectively.

Table 3: Model setting of MLP network.

| Layer Name | Output Size | MLP |
|---|---|---|
| Flatten-1 | 150528 | |
| Linear-2 | 32 | 150528*32 |
| ReLU-3 | 32 | |
| Linear-7 | 200 | 32*200 |

Table 4: Model setting of LeNet.

| Layer Name | Output Size | LeNet |
|---|---|---|
| Conv2d-1 | 12*112*112 | 3,12,5*5,pad=2,stride=2 |
| Sigmoid-2 | 12*112*112 | |
| Conv2d-3 | 12*56*56 | 12,12,5*5,pad=2,stride=2 |
| Sigmoid-4 | 12*56*56 | |
| Conv2d-5 | 12*56*56 | 12,12,5*5,pad=2,stride=1 |
| Sigmoid-6 | 12*56*56 | |
| Linear-7 | 200 | 376322*200 |

## A.3. Supplementary Results

To demonstrate the effectiveness of DLM+ on high-resolution image, we conduct experiments conducted utilizing the ImageNet dataset with image dimensions of (3 * 224 * 224). The result is shown in Fig. 6.

For better viewing the convergence of proposed frameworks under each defense method, we reveal the figures of *PSNR v.s. Iterations* and *Loss v.s. Iterations* as follows. The Figure 7 to Figure 9 are the DLM+ attacks to LeNet under DP with Gaussian noise, Laplacian noise and model sparsification, respectively. Besides, the Figure 11 and Figure 11b are the loss curves of the MLP network under different defenses.

As a supplement to Sections 5.2 and 5.3, Figure 12 shows the variation of loss v.s. iteration for both the LeNet and the MLP network in the learning rate experiment. Similarly, Figure 13 informs the variation of loss v.s. iteration in the local iteration experiment.

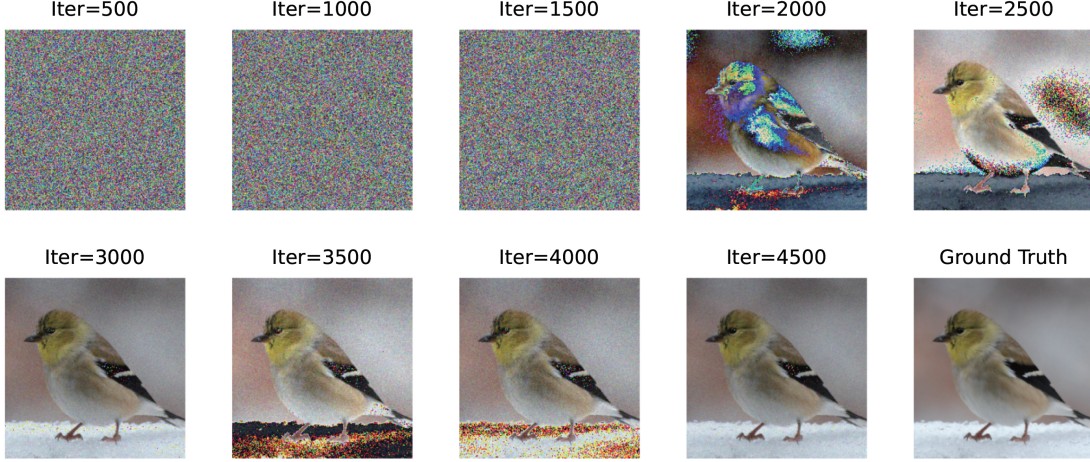

Figure 6: The restoration results of images in ImageNet. At iteration 0 to 1500, the dummy data is a completely random and unrecognizable image. Then the adversary would utilize our frameworks to recover the training data and obtain a clear image after nearly 40 iterations.

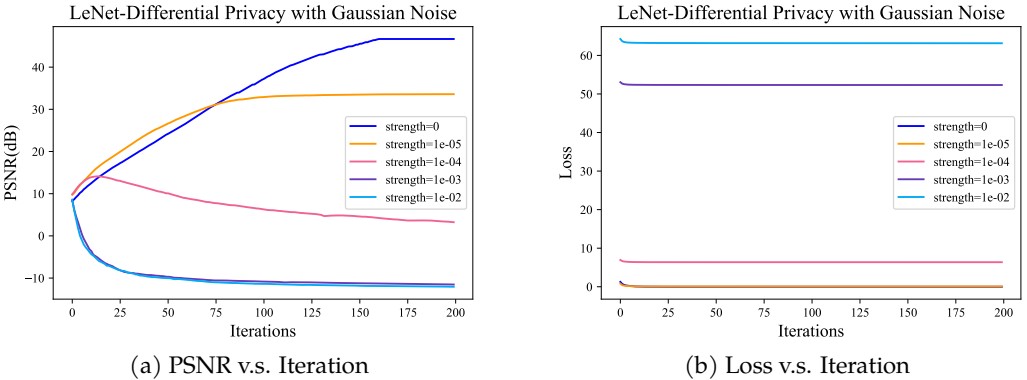

(a) PSNR v.s. Iteration        (b) Loss v.s. Iteration

Figure 7: Performance comparison of applying differential privacy with various Gaussian noise strength to LeNet network.

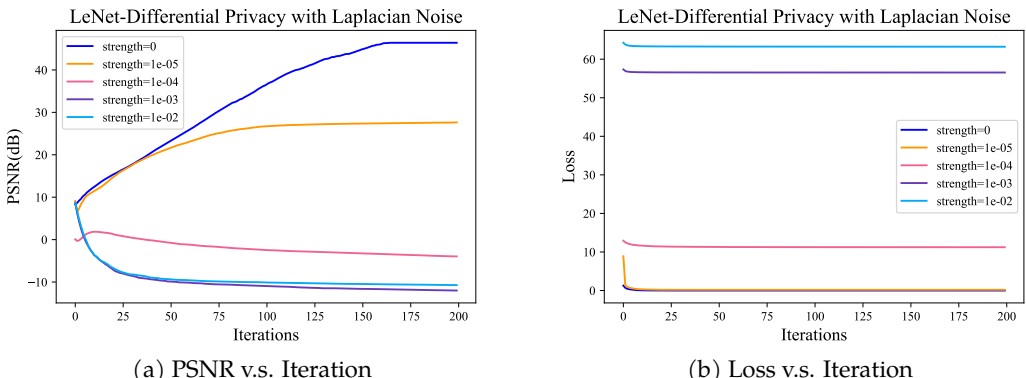

(a) PSNR v.s. Iteration        (b) Loss v.s. Iteration

Figure 8: Performance comparison of applying differential privacy with various Laplacian noise strength to LeNet network.

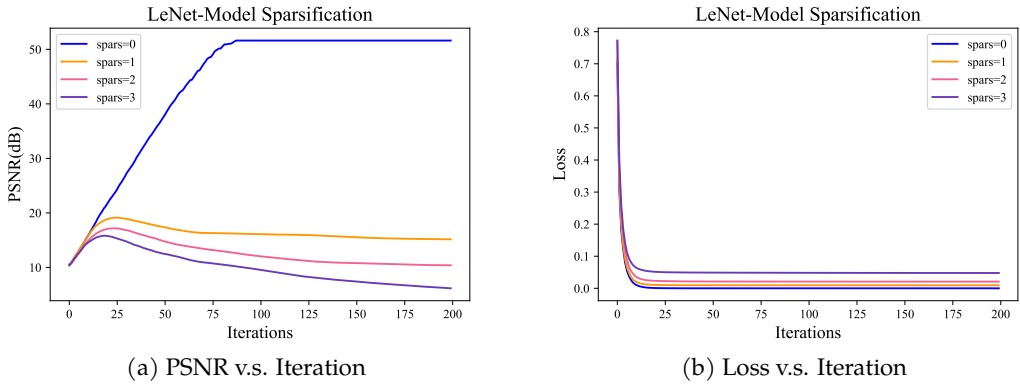

(a) PSNR v.s. Iteration        (b) Loss v.s. Iteration

Figure 9: Performance comparison of LeNet networks with different levels of model sparsity.

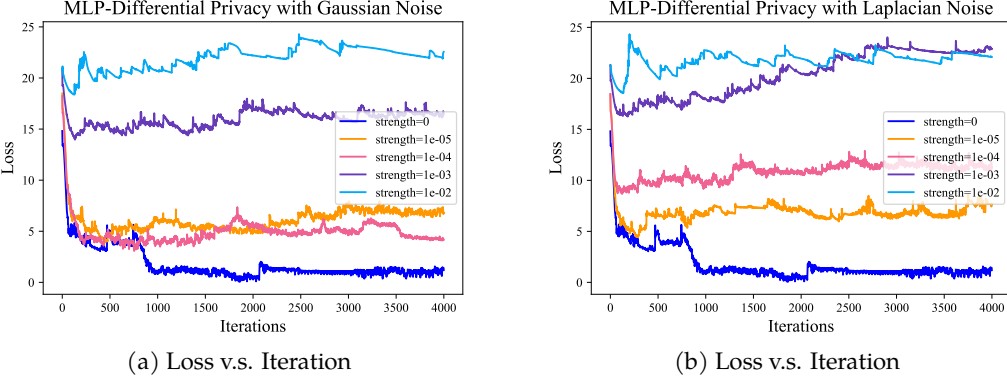

(a) Loss v.s. Iteration                    (b) Loss v.s. Iteration

Figure 10: Performance (Loss) comparison of applying differential privacy with various Gaussian and Laplacian noise strength to MLP network.

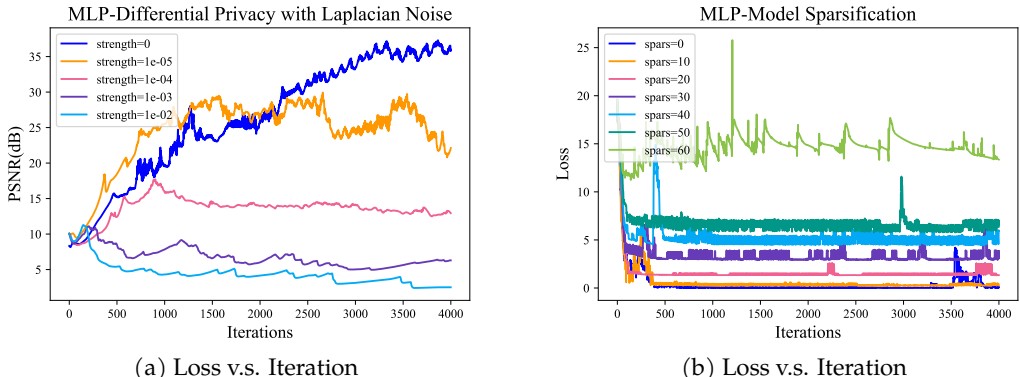

(a) Loss v.s. Iteration                    (b) Loss v.s. Iteration

Figure 11: Performance (Loss) comparison of applying (a) differential privacy with various Laplacian noise strength (b) different levels of model sparsity to MLP network.

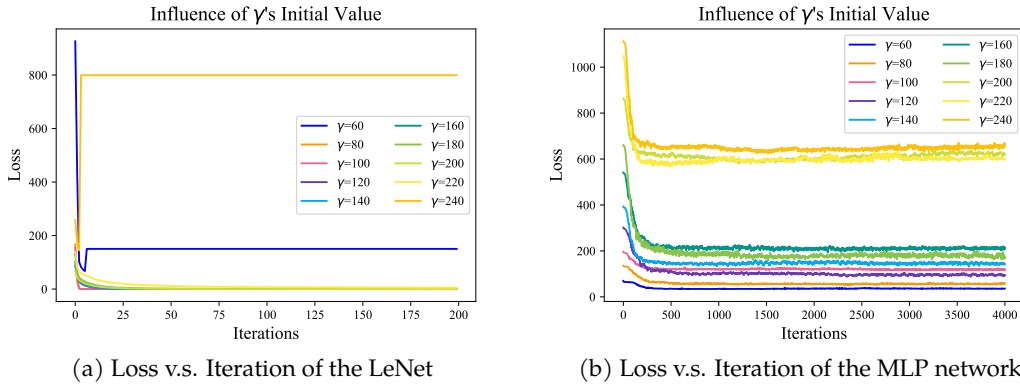

(a) Loss v.s. Iteration of the LeNet       (b) Loss v.s. Iteration of the MLP network

Figure 12: Performance (loss) comparison of both the LeNet and the MLP network with various $\gamma$'s initial value.

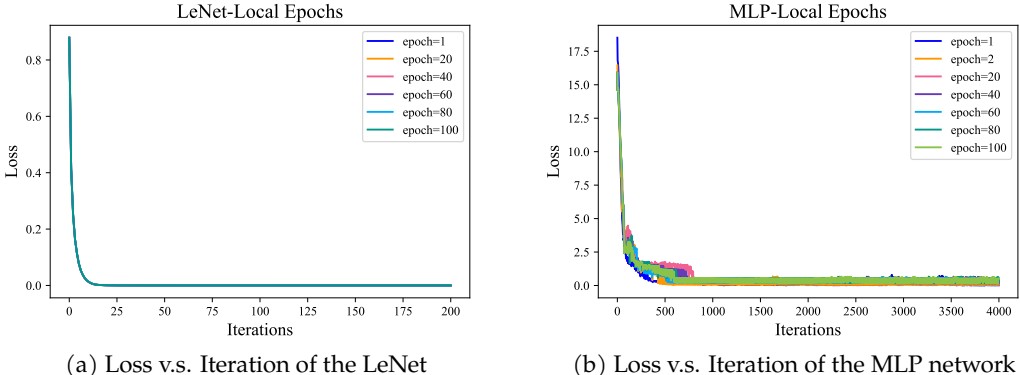

(a) Loss v.s. Iteration of the LeNet  (b) Loss v.s. Iteration of the MLP network

Figure 13: Performance (loss) comparison of both the LeNet and the MLP network with different local epochs.

## B. Batch Data Restoration

Since many researches now focus on the recovery of a mini-batch data, we also use DLM to attack the batch data in this experiment. By following the work of [6], total variance (TV) regularization is added to the objective function DLM+:

$$\mathcal{L}_{DLM+}(\hat{\boldsymbol{x}}, \hat{y}) = \left\| \frac{\nabla_{\boldsymbol{W}_k^t} \mathcal{L}(\hat{\boldsymbol{x}}, \hat{y})}{\|\nabla_{\boldsymbol{W}_k^t} \mathcal{L}(\hat{\boldsymbol{x}}, \hat{y})\|_F} - \frac{\boldsymbol{W}_g^t - \boldsymbol{W}_k^{t+1}}{\|\boldsymbol{W}_g^t - \boldsymbol{W}_k^{t+1}\|_F} \right\|_F^2 + \beta TV(\hat{x}), \tag{13}$$

where the $\beta$ is the scaling factor setting by the adversary.

Subsequently, this new objective function is applied to the DLM+ framework and the result of the four batch-size data recovery is shown in Figure 14. This figure illustrates that our proposed framework is able to restore the batch data as well.

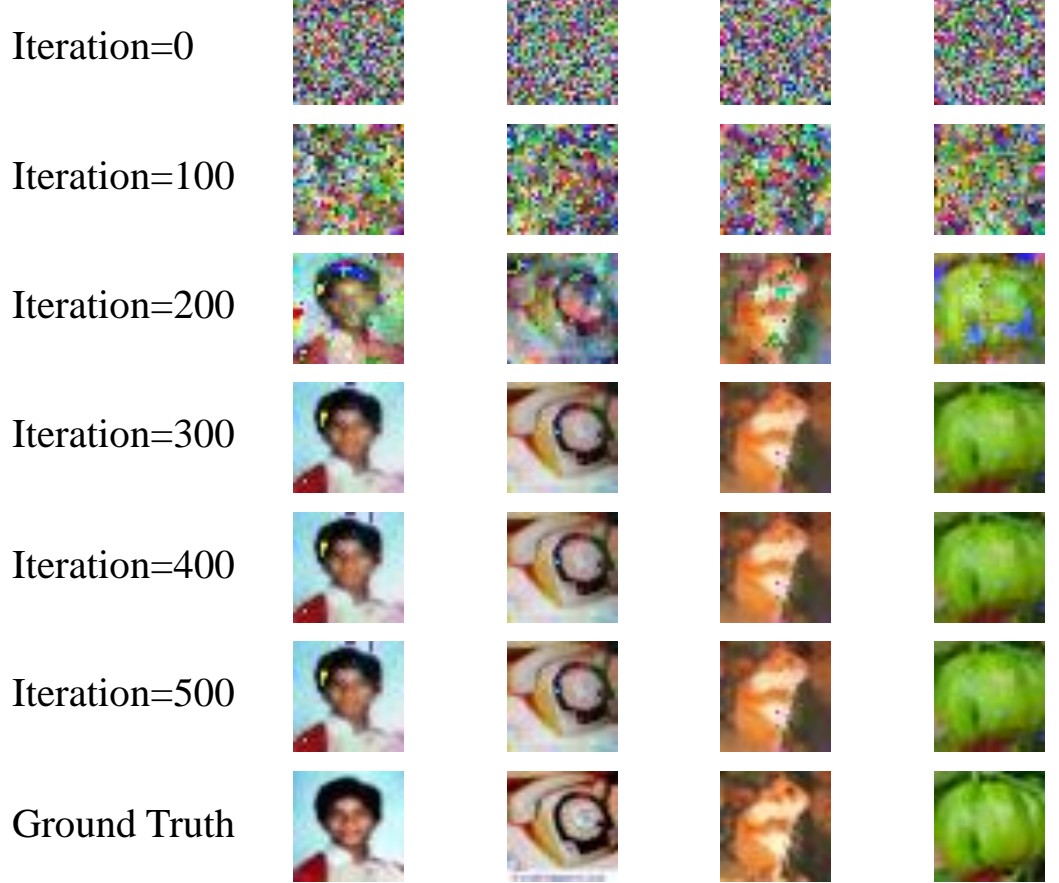

Figure 14: The LeNet restoration result of DLM+ for batch data. Approximately 300 rounds, the ground-truth image has been completely recovered.

