# OpenReview forum: "Deep Leakage from Model in Federated Learning"
_CPAL.cc/2024/Conference — CPAL 2024 (Proceedings Track) Oral_

### Official Review · Reviewer_GcxC · 2023-10-07

**Rating:** 6
**Confidence:** 3

**Review:**

## Overview

The study introduces a novel attack framework, termed DLM and DLM+, that challenges existing federated learning systems where model weights are transmitted rather than the gradient. Unique to DLM(+) is its reliance solely on communicated model parameters and the loss function, making it more flexible than earlier gradient leakage attacks. Comprehensive tests underscore the consistent efficacy of the introduced methods.

## Comments

(+) The advancements brought about by DLM+ are notably superior to other attack baselines.

(+) A series of ablation studies have been carried out, which strongly attest to the efficacy of the proposed method.

(+) This work is well-motivated and proposes a threat to federated learning where only model weights are transmitted.

(-) Referring to line 172, while MNIST, CIFAR10, and CIFAR100 are employed to encompass different image sizes, both sets predominantly contain smaller images (28x28 for MNIST and 32x32 for CIFAR10/100). Integrating datasets with larger images might offer more comprehensive results.

(-) It would be valuable to understand if the presented method can be extended to situations where prior information about the loss function isn't accessible.

---

### Official Review · Reviewer_wxwi · 2023-10-08
**Simple changes to DLG leads to higher attack performance and robustness**

**Rating:** 7
**Confidence:** 3

**Review:**

The paper titled "Deep Leakage from Model Parameters in Federated Learning" investigates the security vulnerabilities within the Federated Learning (FL) framework, primarily focusing on the potential data leakage when transmitting model parameters. The authors highlight the importance of FL in addressing data privacy concerns and the challenges posed by the explosive growth of data. The authors set the stage by mentioning the recent advancements in gradient leakage attacks, which have raised concerns about the security of FL. They emphasize the common requirement for auxiliary information in these attacks, such as model weights, optimizers, and hyperparameters, which can be challenging to obtain in practical scenarios. Furthermore, the paper raises concerns about the security of transmitting model weights in FL, a less explored area. To address these issues, the authors propose two novel attack frameworks, DLM and DLM+, which aim to expose the potential leakage of private client data when transmitting model weights.

To reduce the number of trained variables, the authors propose to approximate learning rate with value of the weight difference of global model divided by gradients. Doing so can get rid of learning rate and gain better performance. Empirical results verify the effectiveness of proposed method. In addition, two defenses against the proposed attacks have been presented.

Pros:

(1) The paper is written clearly and present the main idea and methodology in a very pleasant way. I can directly understand the main mechanism of data leakage after reading the paper once.

(2) The main idea is simple yet effective. A very straight-forward modification leads to significant improvement. I believe the simplicity is a pro rather than con in this case.

(3) Empirical results demonstrate the effectiveness of the proposed method clearly.

(4) The proposed method enjoys appealing robustness to local iteration and optimizer as well.
Cons:

(1) While promising performance is demonstrated, I would like to see the efficiency comparison between DLM+ and DLG. Does the simple change leads to higher computational cost?

---

### Official Review · Reviewer_Gbfu · 2023-10-15
**Review for deep leakage**

**Rating:** 6
**Confidence:** 3

**Review:**

1. Summary:
The paper begins by highlighting the challenges posed by the growth and complexity of data to traditional centralized machine learning. Distributed learning has emerged as a solution, with federated learning (FL) being a notable application. FL aims to protect client data privacy by keeping training data local. However, even without uploading raw training data, FL can still be vulnerable to data leakage.

2. Main Contributions:
a) The paper identifies the possibility of recovering private training data in FL using only transmitted model parameters and loss functions, challenging the foundational security of FL.
b) Two novel attack frameworks, DLM and DLM+, are introduced. These frameworks are applied to FedAvg, a widely-used algorithm in FL. c) The results show that FL architectures that exchange model weights cannot fully protect client data.
d) The paper compares the proposed model leakage attacks with existing gradient leakage attacks, demonstrating the superiority of the new attacks. Additionally, two defenses against these attacks are introduced.

3. Pros:
a) The paper addresses a critical security concern in federated learning, enhancing the understanding of potential vulnerabilities.
b) Introduces two novel attack frameworks, DLM and DLM+, offering a comprehensive approach to understanding data leakage in FL.
c) Provides defenses against the proposed attacks, ensuring a balanced perspective on the issue.

4. Cons:
a) The paper's focus on transmitted model weights might limit its applicability to FL systems that don't rely on this method.
b) The complexity of the proposed frameworks might make them challenging to implement in real-world scenarios.
c) The paper assumes that attackers have access to certain transmitted data, which might not always be the case in secure FL implementations.

---

### Meta-Review · Area_Chair_rwYk · 2023-11-13

**Recommendation:** Accept (Poster)
**Confidence:** 5

**Metareview:**

This paper develops a new attack algorithm for model inversion attacks in federated learning. Different from the majority of prior art, this paper considers the setting where models instead of gradients are being transmitted. All three reviewers agree that this is a novel and interesting work and deserves acceptance to the CPAL conference. During the rebuttal phase, the authors further address the reviewers' minor concerns on this paper's complexity and setting. Therefore, I support the acceptance of this paper.

---

### Decision · Program_Chairs · 2023-11-19

**Decision:**

Accept (Oral)

**Comment:**

This paper develops new attack algorithms, from which private local data might be leaked when model weights are transmitted in the FL framework. Extensive experiments are provided to corroborate the effectiveness of the proposed approaches.

The action PC chair for this paper is Yuejie Chi, who made the decision after carefully reading the paper as well as the comments by all reviewers and AC. The decision is agreed by all PC chairs.